# The Sooner the Better: Investigating Structure of Early Winning Lottery Tickets

## Abstract

The recent success of the lottery ticket hypothesis by Frankle & Carbin (2018) suggests that small, sparsified neural networks can be trained as long as the network is initialized properly. Several follow-up discussions on the initialization of the sparsified model have discovered interesting characteristics such as the necessity of rewinding (Frankle et al. (2019)), importance of sign of the initial weights (Zhou et al. (2019)), and the transferability of the winning lottery tickets (S. Morcos et al. (2019)). In contrast, another essential aspect of the winning ticket, *structure* of the sparsified model, has been little discussed. To find the lottery ticket, unfortunately, all the prior work still relies on computationally expensive iterative pruning.

In this work, we conduct an in-depth investigation of the structure of winning lottery tickets. Interestingly, we discover that there exist many lottery tickets that can achieve equally good accuracy much before the regular training schedule even finishes. We provide insights into the structure of these *early* winning tickets with supporting evidence. 1) Under stochastic gradient descent optimization, lottery ticket emerges when weight magnitude of a model saturates; 2) Pruning before the saturation of a model causes the loss of capability in learning complex patterns, resulting in the accuracy degradation. We employ the memorization capacity analysis to quantitatively confirm it, and further explain why gradual pruning can achieve better accuracy over the one-shot pruning. Based on these insights, we discover the early winning tickets for various ResNet architectures on both CIFAR10 and ImageNet, achieving state-of-the-art accuracy at a high pruning rate without expensive iterative pruning. In the case of ResNet50 on ImageNet, this comes to the winning ticket of $75.31\%$ Top-1 accuracy at $80\%$ pruning rate in only $22\%$ of the total epochs for iterative pruning.

## 1 Introduction

Deep Neural Networks (DNNs) achieve superior accuracy in a wide spectrum of applications through the use of very large and deep models (Goodfellow et al. (2016)). These high-capacity but complex models, however, pose a tremendous challenge for their deployment, particularly in resource-constrained edge environments. Over the years, many techniques have been developed to compress the models to a compact counterpart to alleviate the computational costs. Among these techniques, pruning less important parameters to obtain a compact sub-network has emerged to be a popular and efficient approach (Cun et al. (1990); Han et al. (2015)). In search of these sub-networks, new intuitions are also built up for understanding the DNN working mechanism, one example of which is the recently proposed "lottery ticket hypothesis" by Frankle & Carbin (2018).

The lottery ticket hypothesis states that, once a sub-network is found to match the accuracy of the original neural network, the sub-network (i.e., lottery ticket) together with its initialized weights can be trained in isolation and still achieve accuracy comparable to the original network within a similar number of iterations. This conjecture intrigues discussions on a series of topics, such as the importance of initialization scheme (Liu et al. (2018), Zhou et al. (2019)), the role of over-parameterization in training (Frankle et al. (2019)) and even the transferability of the "winning ticket" (S. Morcos et al. (2019)). However, all these discussions start from the point that the winning ticket has been obtained after painfully long iterative pruning procedures, which often take up to thousands of epochs. When and how a winning ticket can be found in the course of pruning procedures has not been studied; most of the prior works use a traditional way of repeated cycles of pruning and retraining, which makes such study less practical.

In this work, we provide insights to find the winning tickets *early*. We start with an interesting observation that there exist many lottery tickets that can achieve equally good accuracy much before the regular training schedule finishes.

This leads to our in-depth investigation of the weight magnitude; we discover that a model saturates early but not too early under stochastic gradient descent (SGD) optimization. To understand this characteristic of the winning lottery tickets, we conjecture that pruning a premature model causes the loss of capability in learning complex patterns, leading to accuracy degradation. We confirm this conjecture with the empirical evidence as well as the quantitative analysis based on the memorization capacity. Using this analysis framework, we further provide a reasoning behind the success of the gradual pruning over the one-shot pruning. Based on these insights, we identify the early winning tickets for various ResNet architectures on both CIFAR10 and ImageNet, achieving state-of-the-art accuracy at a high pruning rate without expensive iterative pruning. In the case of ResNet50 on ImageNet, this comes to the winning ticket of $75.31\%$ Top-1 accuracy at $80\%$ pruning rate obtained within only $22\%$ of the total epochs for iterative pruning. This promising outcome not only sheds light on understanding the optimization behavior of the pruned models but also enables performance gain for fast training of the pruned models.

## 2 RELATED WORK

The lottery ticket hypothesis was first proposed in Frankle & Carbin (2018) where the presence of a trainable sub-network that achieves compelling final accuracy with the inherited initial values for the un-pruned connections is demonstrated. This paper also argued that the initial weight of the original network is essential for maintaining good accuracy when the model is sparsified. This claim has been extended to the over-parameterized neural networks on the larger datasets in Frankle et al. (2019) with the notion of "rewinding"; the authors claim that rewinding of the weights not to the initial values but to the values after a few epochs can stabilize the accuracy of the winning lottery tickets. As a follow-up work, Zhou et al. (2019) studied the critical components of the lottery tickets such as zeros, signs and the super-mask. Also, S. Morcos et al. (2019) investigated the transferability of winning tickets obtained in one dataset to the network of a similar structure for the other datasets. But none of these focused on the structure of the winning lottery tickets; it involves several repetitions of the full training schedule for iterative pruning and retraining, often taking hundreds to thousands of epochs. In this work, we demonstrate that many winning tickets can be found in the early stage of the baseline training schedule, drastically reducing the computational effort to obtain them.

Most work on the lottery ticket hypothesis, including this work, rely on magnitude-based weight pruning for identifying unimportant weights to be pruned (usually via global sorting). Such an intuitive method was first proposed in Han et al. (2015) and became popular. Afterwards, more complex pruning methodologies have been presented to improve pruning performance, such as having different pruning criteria (Li et al. (2017), Wen et al. (2016), Liu et al. (2017)) or different pruning granularity (Mao et al. (2017), Molchanov et al. (2019)). While these attempts offer insights on training pruned models, there is little discussion about the interesting interplay between the pruning criteria and the structure of a model pruned by it. In this work, we reveal that the pruned structure obtained based on the weight magnitude has tangible impact on the final accuracy, and further propose a way to quantitatively distinguish good structures for pruning.

There have been various strategies to apply pruning to the neural networks. Iterative pruning by Han et al. (2015) involves several repetition of pruning (with gradual increase of the pruning rate) and retraining. One can increase the repetition cycles arbitrarily large to achieve good accuracy at high pruning rate; Frankle & Carbin (2018) employed iterative pruning with hundreds to thousands of pruning epochs to match the baseline accuracy for the challenging neural networks. On the other hand, gradual pruning introduced by Zhu & Gupta (2017) determines the pruning rate and frequency via a polynomial equation as a function of the starting and ending epochs as well as the target pruning rate. Although it provides a systematic pruning schedule, there is lack of discussion about when to start the gradual pruning. Lastly, Lee et al. (2018) proposes a method of pruning weights at initialization. This method takes most advantage in performance since a pruned model can be obtained without any expensive retraining procedures. However, its effectiveness has not been demonstrated for the challenging neural networks on large datasets such as ImageNet. In contrast, we propose a mechanism to identify the winning lottery ticket early in the course of baseline training so that we can avoid costly iterative pruning while maintaining the baseline accuracy.

## 3 STRUCTURE OF EARLY WINNING TICKETS

In this section, we extend the lottery ticket hypothesis by Frankle & Carbin (2018); Frankle et al. (2019) to discuss the early winning tickets. The lottery ticket hypothesis can be summarized as:

| v \ s | 10 | 20 | 40 | 60 | 80 | 100 | 120 | 140 | 160 | 180 | 200 |
|---|---|---|---|---|---|---|---|---|---|---|---|
| Baseline | 80.26 | 83.74 | 82.72 | 85.30 | 84.31 | 84.95 | 87.55 | 92.08 | 91.63 | 92.11 | 92.05 |
| 0 | 89.81 | 90.15 | 89.97 | 90.50 | 89.89 | 90.08 | 89.94 | 89.72 | 90.10 | 89.67 | 89.37 |
| 5 | 89.83 | 90.30 | 90.91 | 91.08 | 91.28 | 91.65 | 91.52 | 91.50 | 91.48 | 91.42 | 91.13 |
| 10 | 90.21 | 90.31 | 90.74 | 91.12 | 91.37 | 91.55 | 91.43 | 91.43 | 91.48 | 91.45 | 91.32 |
| 20 | 90.32 | 90.27 | 91.05 | 91.13 | 91.39 | 91.47 | 91.25 | 91.43 | 91.33 | 91.25 | 91.45 |
| 40 | 90.43 | 90.46 | 90.63 | 90.97 | 91.58 | 91.67 | 91.36 | 91.56 | 91.54 | 91.43 | 91.59 |
| 60 | 90.03 | 90.19 | 90.82 | 90.96 | 91.45 | 91.56 | 91.30 | 91.64 | 91.67 | 91.38 | 91.45 |
| 80 | 90.05 | 90.28 | 90.67 | 91.18 | 91.21 | 91.42 | 91.34 | 91.62 | 91.54 | 91.23 | 91.45 |
| 100 | 90.55 | 90.26 | 90.74 | 91.39 | 91.36 | 91.54 | 91.43 | 91.50 | 91.34 | 91.39 | 91.54 |
| 120 | 90.17 | 90.55 | 90.94 | 91.31 | 91.35 | 91.49 | 91.62 | 91.55 | 91.32 | 91.38 | 91.41 |
| 140 | 90.16 | 90.50 | 90.88 | 91.11 | 91.46 | 91.77 | 91.14 | 91.65 | 91.41 | 91.48 | 91.25 |
| 160 | 90.16 | 90.29 | 90.82 | 91.30 | 91.34 | 91.67 | 91.41 | 91.56 | 91.36 | 91.40 | 91.44 |
| 180 | 90.73 | 90.33 | 90.80 | 91.28 | 91.56 | 91.60 | 91.58 | 91.66 | 91.31 | 91.49 | 91.38 |
| 200 | 90.33 | 90.64 | 90.86 | 90.94 | 91.47 | 91.68 | 91.37 | 91.55 | 91.55 | 91.15 | 91.32 |

Figure 1: Extended lottery ticket experiments using ResNet20 on CIFAR10. From the baseline ResNet20 training, weight-magnitude based pruning is applied at different epoch (columns, labeled as s) to obtain the sparsified structure. Each structure is then rewound to the weights from another epoch (rows, labeled as v), where v=0 indicates rewind to the initial weight. Lottery tickets emerge much earlier before the full training ends, achieving matching accuracy compared to the conventional winning ticket, i.e., (s=200,v=200).

for a given network of $f(x; w_0 \odot m_0)$ with the initial weight $w_0$ and the mask of all ones $m_0$, there exists a winning lottery ticket $m_f$ where $|m_f|/|m_0| = 1 - p\%$ ($p$ is the pruning rate) and training of $f(x; w_v \odot m_f)$ for $0 < v \ll f$ achieves test accuracy comparable to the baseline $f(x; w_f \odot m_0)$.

There are two main components of a winning lottery ticket: the sparsified structure $m_f$ and the weight that initializes it, $w_v$. In the previous work, $m_f$ has been obtained only after expensive iterative pruning. We characterize the structure of the early lottery ticket $m_s$ where $s \ll f$, then propose a strategy for finding it early.

### 3.1 EXTENDED LOTTERY TICKET EXPLORATION

Frankle et al. (2019) suggests that a lottery ticket found after a baseline training can achieve the accuracy of the baseline model if it is initialized with the weight of the baseline model after a few epochs, which is called *rewinding*. We extend this exploration toward different structures obtained at different epochs of the baseline training (via magnitude-based weight pruning).

Fig. 1 shows the validation accuracy after retraining of a ResNet20 model on CIFAR10 pruned at different epochs of its baseline training. The same learning rate schedule of 0.1 reduced by 10x at epoch 120 and 160 is used for both the baseline training and retraining, and the total number of epochs is 200. The rows (= $v$) correspond to the different rewinding epoch, whereas the columns (= $s$) correspond to the different epoch that we apply one-shot pruning (pruning rate= 80%). For example, the validation accuracy along the lottery ticket configurations of ($s = 200, v = 0 \sim 200$) resembles the phenomena of "rewinding" observed in Frankle et al. (2019).

Interestingly, Fig. 1 further demonstrates that the winning tickets emerge at much earlier epochs of the baseline training; the lottery ticket configurations of ($s \geq 100, v \geq 5$) achieve almost the same accuracy as the accuracy of ($s = 200, v = 200$). The accuracy then gradually decreases as $s \leq 80$. This result implies two important aspects: 1) a winning ticket can be found in the middle of the baseline training so that one can avoid expensive iterative pruning used in the prior work, and 2) the winning ticket, however, does not emerge arbitrarily early in the process of training. In the following sections, we investigate the characteristics of these *early* winning tickets. In particular, we focus on their structure, as the weight initialization is not the major factor provided a proper rewinding.

### 3.2 ANALYSIS ON WEIGHT MAGNITUDE

As the first step of understanding the characteristics of the early winning tickets, we focus on the important quantity of pruning, *weight magnitude*. At pruning, we determine the structure of the sparsified model based on the rank of the

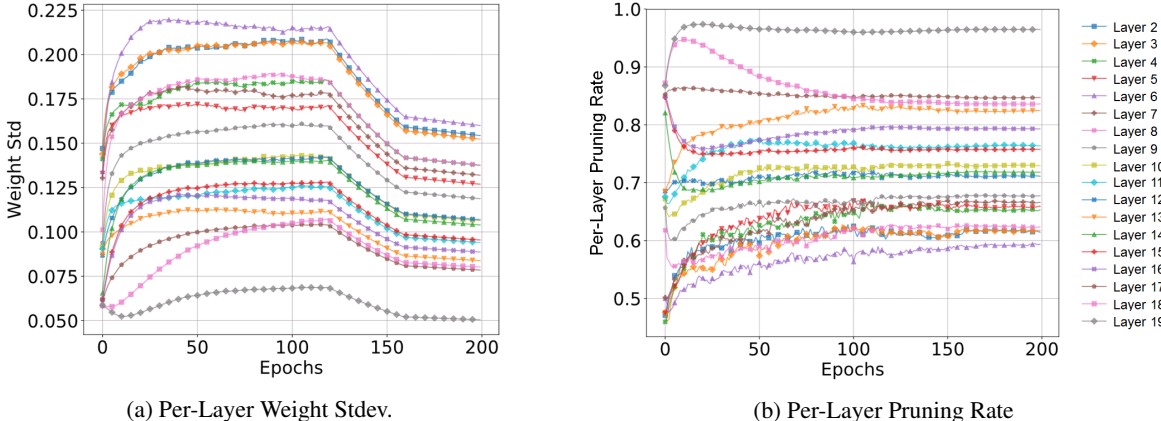

(a) Per-Layer Weight Stdev.                (b) Per-Layer Pruning Rate

Figure 2: (a) Change of the standard deviation of weights ($W_{std}$) in each layer during the training of ResNet20 on CIFAR10. In the beginning of the training, the weights of different layers change in different rate, dominated by the gradient terms. When the training evolves, weight magnitude is primarily determined by the interplay between learning rate and weight decay, resulting in parallel movement of $W_{std}$. (b) Change of per-layer pruning rate over the epochs. Due to regular shift of $W_{std}$ at the later epochs of training, the per-layer pruning rate converges to a saturating point, emerging stable structure for pruning.

weight magnitude (via global sorting). Therefore, the change in the weight magnitude during training has large impact on the lottery ticket structure.

In Fig. 2a, we measure the standard deviation of weight ($W_{std}$) for each layer in the course of training ResNet20 on CIFAR10 (note: the mean of weight is typically near zero). The same learning rate schedule is used as above. The first thing to note is that the change in $W_{std}$ has strong correlation with the learning rate change. In particular, different layers show different rate of change in $W_{std}$ when the learning rate is 0.1, but from the second learning rate (after epoch 120), all the weights follow a very similar decreasing trend.

This trend in $W_{std}$ can be understood via steps of stochastic gradient descent. From the typical setting of weight update with momentum and weight decay, we have:

$$v_{t+1} = mv_t + (\lambda w_t + w_{g,t}), \; w_{t+1} = w_t - \eta v_{t+1}. \tag{1}$$

where $w_t$, $w_{g,t}$ and $v_t$ are the weight, gradient and momentum at step $t$, respectively; $m$ is the momentum factor, $\eta$ is the learning rate and $\lambda$ is the weight decay factor. After $n$ steps,

$$v_{t+n} = m^n v_t + \sum_{k=1}^{n}(m^{n-k}w_{g,t+k-1}) + \lambda \sum_{k=1}^{n}(m^{n-k}w_{t+k-1}), \tag{2}$$

$$w_{t+n} = (1 - \eta\lambda)^n w_t - \eta \sum_{k=1}^{n}((1 - \eta\lambda)^{n-k}(mv_{t+k-1} + w_{g,t+k-1})). \tag{3}$$

From this derivation, we can see that the two factors determine the mode of change in $W_{std}$. When learning rate is high, the gradient terms play the major role in weight update. On the other hand, if the gradient activity becomes low, e.g., when the learning rate is low and the gradients oscillate around zero, we can further simplify Eq. 3. Assume that $m^n v_t$ approaches to zero when $n$ is relatively large, $w_{t+k-1} \approx w_t$, and $\sum_{k=1}^{n}((1 - \eta\lambda)^{n-k}w_{g,t+k-1})$ and $\sum_{k=1}^{n}(m^{n-k}w_{g,t+k-1})$ approach to zero as the gradients oscillate around zero, we have $v_{t+i} \approx \frac{\lambda w_t}{1-m}$. Then $w_{t+n}$ is approximated as,

$$w_{t+n} \approx (1 - \eta\lambda)^n w_t - \eta \sum_{i=0}^{n-1}((1 - \eta\lambda)^i \frac{m\lambda w_t}{1 - m}). \tag{4}$$

Note that $(1 - \eta\lambda)^n \approx 1 - n\eta\lambda$ since $\eta\lambda \ll 1$ and $\sum_{i=0}^{n-1}((1 - \eta\lambda)^i) \approx n$. Thus,

$$w_{t+n} \approx (1 - n\eta\lambda)w_t - \frac{n\eta\lambda m w_t}{1 - m} = (1 - \frac{n\eta\lambda}{1 - m})w_t. \tag{5}$$

In other words, when the gradient activity is low, the change in $W_{std}$ is dominated by the both the learning rate and the weight decay. As an example, in Fig. 2a, $W_{std}$ is decreased with the slope following Eq. 5 when learning rate is low; $W_{std}$ of layer 10 is decreased by 1.5e-3 in 782 updates of CIFAR10 (with $\lambda =$ 1e-4 and $m = 0.9$), confirming the slope from Eq. 5.

These two modes governing the change of weight magnitude are critical for understanding the behavior of pruning. In particular, the interplay between learning rate and weight decay causes the per-layer pruning rate to converge after enough number of epochs. Fig. 2b shows the *pseudo* per-layer pruning rate, where we just measure the layer-wise pruning rate without really pruning out the weights in the model, for the same CIFAR10 training experiment. As the figure shows, the per-layer pruning rates saturate around epoch 100, indicating that pruning before that would select the weights based on the premature model.

Moreover, although the weight magnitude is an important factor, we discovered that the ranking of individual weight does not play a critical role in winning lottery tickets. Specifically, we empirically show that there exist many winning lottery tickets that are vastly different in terms of its sparse structure $m_v$. Fig. 3 shows the hamming distance of the sparse structure of the lottery tickets at different configuration $(s, v)$. Note that the two distant lottery tickets (e.g., $(s = 200, v = 100)$ and $(s = 100, v = 100)$) show large hamming distance of 0.159 (where $x/y = 200/100$) while achieving the equally good accuracy as shown in Fig. 1, indicating that the ranking of the weights itself can not explain the quality of the structure of the lottery ticket.

| x \ y | 100 | 120 | 140 | 160 | 180 | 200 |
|-------|-----|-----|-----|-----|-----|-----|
| 100 | 0 | 0.154 | 0.153 | 0.158 | 0.159 | 0.159 |
| 120 | | 0 | 0.052 | 0.079 | 0.080 | 0.082 |
| 140 | | | 0 | 0.047 | 0.047 | 0.048 |
| 160 | | | | 0 | 0.008 | 0.011 |
| 180 | | | | | 0 | 0.006 |
| 200 | | | | | | 0 |

Figure 3: The hamming distance of the sparse structure of the lottery tickets at different configuration for ResNet20 on CIFAR10. $x/y$ denotes the distance between $(s = x)$ and $(s = y)$. A large distance between two lottery tickets with equally good accuracy suggests the existence of many winning tickets.

### 3.3 Understanding Impact of Pruned Structure

The weight magnitude analysis motivates us not to prune a model too early. It is also implied that a distance-based metric might not reveal the winning structure of the model at different epochs. To understand the early winning structures, we further investigate the impact of the sparsified structure on the final accuracy. Considering that a highly pruned network is likely to have limited learning capability, we make a conjecture that the accuracy degradation of a pruned model is due to the loss of capability for learning complex pattern if pruned too early. Recently, Li et al. (2019) reveals that the training behavior of a sufficiently over-parameterized model with non-linearity highly depends on the learning rate schedule, where a model tends to memorize the complex patterns when a small learning rate is applied while learning simple patterns with a large learning rate. To validate this claim in the context of pruning, we construct an experiment where a model is *pseudo*-pruned at every epoch of the baseline training then retrained with large or small learning rate for just one epoch. For the pseudo-pruned-then-retrained (PPR) model at each baseline epoch, we measure the validation accuracy recovered from the retraining.

Fig.4 shows the result of this experiment on CIFAR10 ResNet20. When it is retrained with the large retraining learning rate ($= 0.1$), in just 1-epoch retraining, the PPR models from all the baseline epochs achieve the accuracy matching with the baseline accuracy. This indicates that those pruned models maintain the capability of learning the simple patterns. Whereas, when the small retraining learning rate ($= 0.01$) is used, the accuracy of the PPR models pruned at later epochs (epoch 100-200) is higher than the accuracy of the models from earlier epochs (epoch 20-60). This reveals that the PPR models from different epoch exhibit varying capability of learning complex patterns. In particular, the accuracy of the PPR models increases until around 100 epoch of the baseline training, then it saturates. Note that this coincides with the epoch when the early winning tickets emerges in Fig. 1.

Based on this observation, we hypothesize that the models pruned at 100 epoch of the baseline training or later will preserve the capability of learning complex patterns. To validate this hypothesis, we conducted the memorization test proposed by Boo et al. (2019), where a model is trained with training data of varying size with the randomized labels.

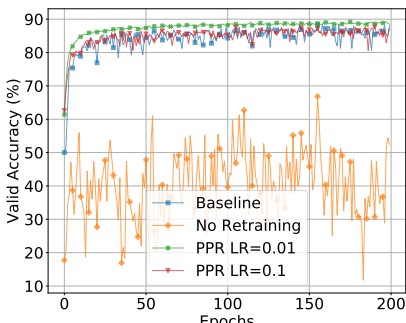

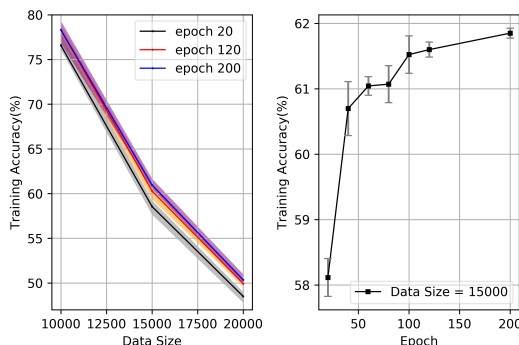

Figure 4: The pseudo-pruned-then-retrained (PPR) models for CIFAR10 ResNet20, which exhibit increasing capability of memorizing the complex patterns over the epochs. Validation accuracy for the baseline training and validation accuracy after pruning without retraining are also included for comparison.

Figure 5: Left: Memorization capacity (i.e., training accuracy) of the models pruned at 3 different epochs. The models pruned at epoch 120 and 200 show almost identical memorization capacity, whereas the model pruned at epoch 20 exhibits lower memorization capacity. Right: Memorization capacity of models pruned at different epochs (training data size = 15000).

A model has "high memorization capacity" if it achieves high training accuracy for a large data size. Fig. 5 (left) shows the training accuracy of the models pruned at different epochs, measured from 10 independent simulations (the average capacity shown as solid lines, and $min$ and $max$ shown as the shaded regions). As the training data size increases, the memorization capacity decreases. The model pruned too early (i.e., at epoch 20) suffers higher degradation in the memorization capacity compared to the models pruned at later epochs (i.e., at epochs 120 or 200). Also, Fig. 5 (right) shows that those models pruned at later than 100 epochs (i.e., the early winning tickets) exhibit similar memorization capacity. This result not only confirms our conjecture on the impact of pruned structure to the capability of learning complex patterns, but also leads us to employ the PPR accuracy check as a computationally reasonable heuristic to discover the early winning tickets.

### 3.4 Understanding Gradual Pruning

Gradual pruning is a popular pruning approach that applies pruning gradually over a period of training. For example, gradual pruning proposed by Zhu & Gupta (2017) provides a systematic way to schedule iterative pruning as follows:

$$s_t = s_f + (s_i - s_f) \left( 1 - \frac{t - t_0}{n \Delta t} \right)^3 , \tag{6}$$

where $s_f$ and $s_i$ are the final and initial sparsity, and $t$ is the time when pruning is applied. Eq. 6 determines how much sparsity is applied at a certain time step $t$. But it is still a user-hyper-parameter to decide when $t_0$ or how often $\Delta t$ apply pruning. Based on the insights we discussed in the previous section, we explain why gradual pruning helps to obtain better lottery tickets.

The reasoning behind the gradual pruning is that the model can be changed graciously if the pruning is applied gradually. In terms of the early winning ticket analysis, there are two factors playing the critical role: 1) by applying low pruning rate in the beginning, the structure found at that pruning level can preserve the memorization capacity better, 2) once pruning is applied, the remaining weights of the pruned model is updated via SGD, granting a chance for the pruned model to adopt its weights toward better accuracy. Thanks to these two factors, a structure with better memorization capacity can be found when the increased pruning rate is applied next time.

Table 1 confirms this explanation using the CIFAR10 ResNet20 example. In this experiment, we perform the memorization test for the 4-step gradual pruning as well as four 1-step pruning at the corresponding epoch for comparison. The gap in memorization capacity is maintained across the different epochs, demonstrating that the memorization capacity is maintained thanks to the gradual application of pruning and the evolution of weights after pruning. This suggests a strategy for gradual pruning where 1) we can use gradual pruning to reduce the loss of capability in learning complex patterns, and 2) by applying a smaller pruning rate in the beginning, we can start pruning early and finding the winning lottery tickets faster. The benefit of this strategy combining gradual pruning with the early winning tickets will be demonstrated in Sec. 4.

Table 1: Memorization capacity (i.e., train accuracy) comparison between 1-shot and 4-shot pruning.

| Epoch | 120 | 160 | 200 | 240 | Unit |
|---|---|---|---|---|---|
| Pruning rate | 40 | 60 | 70 | 80 | % |
| 1-shot pruning | 79.72 | 66.88 | 59.81 | 50.46 | % |
| 4-shot gradual pruning | 79.72 | 68.83 | 61.21 | 51.45 | % |

## 4 EXPERIMENTS: FINDING WINNING TICKET EARLY

In this section, we demonstrate our strategy of finding the early winning tickets over popular neural networks on CIFAR10 and ImageNet. The detail experimental setup is described in Appendix A. We perform the lottery ticket experiments of Sec. 3.1 for both one-shot and gradual pruning with the pruning rate of $80\%$. We also conduct the PPR accuracy check of Sec. 3.3 to predict from which epoch the early winning ticket can be found. By comparing the two results, we demonstrate that the proposed heuristic for finding early winning tickets works robustly across the networks and the datasets. Furthermore, we showcase our gradual pruning strategy by comparing the performance in terms of the accuracy and the required pruning epochs with the existing lottery ticket approaches.

### 4.1 EXPERIMENTS ON CIFAR10

Table 2 summarizes the lottery ticket experiments of ResNet20 and ResNet56 on CIFAR10 dataset. In case of one-shot pruning, the winning tickets can be found from epoch 100. In case of gradual pruning, the gradual pruning schedule can start from epoch 75 (which is earlier than the one-shot pruning). Note that the gradual pruning can achieve better accuracy as it can preserve higher memorization capacity as discussed in Sec. 3.4. Fig. 6a shows the pseudo-pruning curve that highlights the presence of the winning tickets from the epoch around 100, which is consistent with the results of the lottery ticket experiment in Table 2.

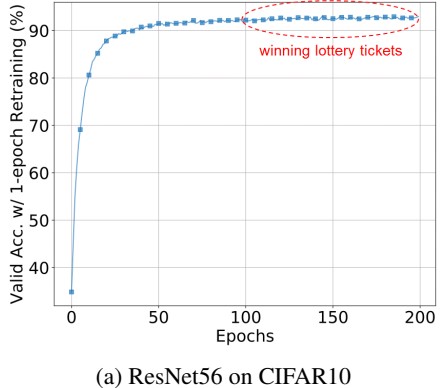

(a) ResNet56 on CIFAR10

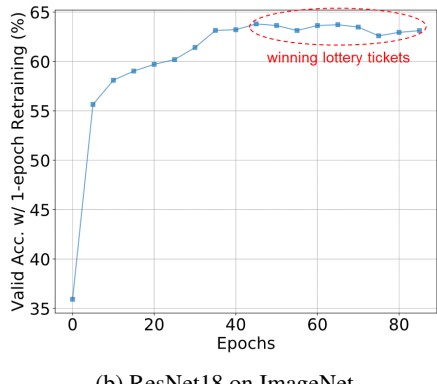

(b) ResNet18 on ImageNet

Figure 6: The results of PPR accuracy check (retraining learning rate=0.01) for (a) ResNet56 on CIFAR10 (sampled at every epoch) and (b) ResNet18 on ImageNet (sampled every 5 epochs).

### 4.2 EXPERIMENTS ON IMAGENET

Table 3 summarizes the lottery ticket experiments on ImageNet dataset, and the predicted results of the early winning tickets from the pseudo-pruning are shown in Fig. 6b. Similar to the CIFAR10 experiments, the results of the lottery ticket experiment matches with the results from the PPR accuracy check (which indicates epoch 45 for early winning tickets), demonstrating the robust behavior of the proposed strategy of finding the early winning tickets. Furthermore, the gradual pruning at the early winning tickets achieves the accuracy near to the baseline (ResNet18: baseline=$69.7\%$ vs ours=$69.24\%$, ResNet50: baseline=$75.7\%$ vs ours=$75.31\%$), showcasing the superior quality of the winning tickets discovered by the proposed gradual pruning strategy.

Table 2: One-shot and gradual pruning at different epochs for ResNet20 and ResNet56 on CIFAR10. To clarify randomness effect, the average accuracy (%) over 10 different runs is reported.

| One-Shot | 20 | 60 | 100 | 120 | 160 | 200 |
|---|---|---|---|---|---|---|
| ResNet20 | 90.01 ($\pm$0.07) | 91.20 ($\pm$0.28) | **91.49** ($\pm$0.27) | **91.37** ($\pm$0.24) | **91.63** ($\pm$0.19) | **91.63** ($\pm$0.21) |
| ResNet56 | 92.94 ($\pm$0.28) | 93.60 ($\pm$0.18) | **94.07** ($\pm$0.16) | **94.24** ($\pm$0.19) | **94.29** ($\pm$0.17) | **94.34** ($\pm$0.14) |
| Gradual | 25-75 | 50-100 | 75-125 | 100-150 | 125-175 | 150-200 |
| ResNet20 | 91.43 ($\pm$0.37) | 91.56 ($\pm$0.28) | **91.78** ($\pm$0.26) | **91.87** ($\pm$0.17) | **91.84** ($\pm$0.32) | **91.83** ($\pm$0.08) |
| ResNet56 | 93.34 ($\pm$0.19) | 93.89 ($\pm$0.12) | **94.03** ($\pm$0.29) | **94.25** ($\pm$0.19) | **94.15** ($\pm$0.17) | **94.24** ($\pm$0.27) |

Table 3: One-shot and gradual pruning for ResNet18 and ResNet50 on ImageNet.

| One-Shot Pruning | | 0 | 15 | 30 | 45 | 60 | 75 | 90 |
|---|---|---|---|---|---|---|---|---|
| ResNet18 | Top-1 Acc. (%) | 64.89 | 66.53 | 67.30 | **68.32** | **68.79** | **68.98** | **69.03** |
| ResNet50 | Top-1 Acc. (%) | 71.65 | 73.11 | 73.71 | **74.51** | **74.90** | **74.92** | **75.03** |
| Gradual Pruning | | 0-30 | 10-40 | 20-50 | 30-60 | 40-70 | 50-80 | 60-90 |
| ResNet18 | Top-1 Acc. (%) | 66.11 | 67.31 | 68.31 | 68.64 | **69.24** | **69.06** | **69.18** |
| ResNet50 | Top-1 Acc. (%) | 72.90 | 73.72 | 74.54 | 74.92 | **75.26** | **75.31** | **74.94** |

### 4.3 PERFORMANCE GAIN FROM EARLY WINNING TICKET

To demonstrate the performance gain from our gradual pruning strategy, the pruning results of the proposed algorithm (GP + EWT) and the previous implementation of iterative pruning (IP) along with the winning ticket (WT) of Frankle & Carbin (2018) are shown for the ResNet variants on CIFAR10 and ImageNet in Table 4. Our GP+EWT algorithm consistently achieves high pruning rate with the number of pruning epochs even lower than the regular retraining epochs for all the models, i.e., $80\%$ pruning with negligible accuracy degradation on ResNet50 for ImageNet. In contrast, the iterative pruning approach (IP+WT) achieves the similar accuracy at the cost of more than $4.5\times$ increase in the total training epochs.

## 5 CONCLUSION

In this work, we investigate the structure of the winning lottery ticket, which leads to the computationally efficient discovery of the winning lottery tickets. Based on a careful analysis of the characteristics of the structure of the winning lottery tickets, we proposed a computationally reasonable heuristic to identify when the early lottery tickets emerge. Furthermore, we proposed a gradual pruning strategy incorporating the early lottery ticket analysis to achieve high accuracy at large pruning rate. This results in the state-of-the-art accuracy on ResNet50 for $80\%$ pruning only within $22\%$ of the total epochs for iterative pruning.

Table 4: Performance gain by our gradual pruning strategy on CIFAR-10 and ImageNet.

| | ResNet20 (CIFAR-10) $PR^*$ / acc.$\Delta^\dagger$ / epochs | ResNet18 (ImageNet) $PR$ / acc.$\Delta$ / epochs | ResNet50 (ImageNet) $PR$ / acc.$\Delta$ / epoch |
|---|---|---|---|
| GP + EWT | 80.0 / -0.20 / **150+200**$^{**}$ | 80.0 / -0.56 / 70+90 | **80.0 / -0.4 / 80+90** |
| IP + WT Frankle et al. (2019) | **82.2** / 0.00 / 1600 | - / - / - | 79.0 / 0.0 / 720 |

GP: gradual pruning, IP: iterative method, EWT: early winning ticket, WT: winning ticket
$^*$: pruning rate (PR). $^{**}$: number of pruning epochs + regular retraining epochs
$^\dagger$: the delta of accuracy is measured against the baseline accuracy. **Bold**: highlight of comparison

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

Table 5: Memorization capacity comparison for different group size.

| Group size | 1 | 2 | 4 | 8 | 16 | 32 | 64 |
|---|---|---|---|---|---|---|---|
| Mem. Cap | 60.3 | 60.4 | 59.4 | 59.1 | 58.5 | 55.0 | 52.4 |

# A    EXPERIMENTAL SETUP

## A.1    EXPERIMENTS ON CIFAR10

We try to find the winning tickets early for ResNet20 (baseline accuracy: 92.5%) and ResNet56 (baseline accuracy: 94.25%) on CIFAR10 dataset. All the baseline models are trained using SGD optimizer with momentum ($m =$0.9) and weight decay ($\lambda =$1e-4) for 200 epochs. The initial learning rate for the first 120 epochs is 0.1 and decrease to 0.1$\times$ every 40 epochs. For the lottery ticket experiments, the retraining schedule is the same as the baseline learning rate schedule. Following convention, we do not prune the first and the last layers of the ResNets.

## A.2    EXPERIMENTS ON IMAGENET

We further try to find the winning tickets early for ResNet18 (baseline accuracy: 69.7%) and ResNet50 (baseline accuracy: 75.7%) on ImageNet dataset. All the baseline models are trained using SGD optimizer with momentum ($m =$0.9) and weight decay ($\lambda =$1e-4) for 90 epochs. The initial learning rate for the first 30 epochs is 0.1 and decrease to 0.1$\times$ every 30 epochs. For the lottery ticket experiments, the retraining schedule is the same as the baseline learning rate schedule. Following convention, we do not prune the first and the last layers of the ResNets.

**Weight Tensor**

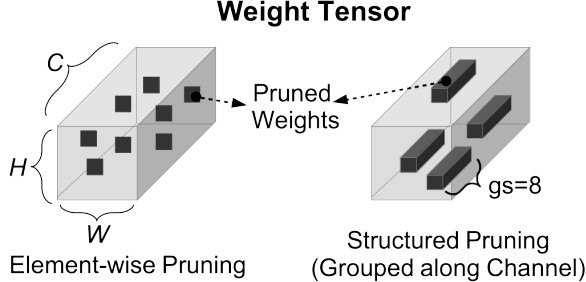

Figure 7: Structured pruning with variable group size. We consider a group of weights along the channel dimension with a varying group-size (gs). In case of gs=1, it is the same as the element-wise pruning.

# B    DISCUSSION

## B.1    IMPACT OF STRUCTURED SPARSITY ON LOTTERY TICKET HYPOTHESIS

In this section, we expand our scope of analysis on the characteristics of the early winning tickets toward the sparsity obtained via structured pruning. Structured pruning is a popular method that prunes weights by a group. For example, Liu et al. (2017) proposes channel-pruning, where a group consists of weights along each channel. The main motivation of looking at the structured sparsity is that there is a significant discrepancy in the best pruning rate we can achieve between the element-wise pruning (i.e., group-size=1) and the structured pruning (i.e., group-size=channel). For example, on CIFAR10-VGG19, the element-wise pruning achieves 95% of pruning rate whereas the channel-pruning achieves 70% of pruning rate for the same level of accuracy (Liu et al. (2018)). Although such discrepancy has been observed for quite a while, there is no in-depth investigation to understand why. To find out the reasoning behind it, we analyze the impact of structured sparsity with varying group sizes in the context of the lottery ticket hypothesis, as illustrated in Fig.7.

First, we revisit the extended lottery ticket experiments of Fig. 1 while applying group sparsity along the channel dimension (instead of pruning individual weights). Fig. 8 shows the results when the group size ($gs$) is 8. Comparing

| v\s | 10 | 20 | 40 | 60 | 80 | 100 | 120 | 140 | 160 | 180 | 200 |
|---|---|---|---|---|---|---|---|---|---|---|---|
| Baseline | 80.26 | 83.74 | 82.72 | 85.30 | 84.31 | 84.95 | 87.55 | 92.08 | 91.63 | 92.11 | 92.05 |
| 0 | 90.10 | 90.27 | 89.92 | 89.85 | 90.02 | 89.73 | 90.07 | 89.76 | 90.19 | 89.95 | 89.85 |
| 5 | 90.12 | 90.14 | 89.99 | 90.65 | 90.41 | 90.42 | 90.46 | 90.29 | 90.49 | 90.17 | 90.53 |
| 10 | 90.04 | 90.30 | 90.20 | 90.37 | 90.43 | 90.58 | 90.68 | 90.41 | 90.41 | 90.68 | 90.46 |
| 20 | 90.35 | 90.43 | 90.23 | 90.56 | 90.30 | 90.65 | 91.02 | 90.88 | 90.65 | 90.29 | 90.58 |
| 40 | 90.30 | 90.38 | 90.05 | 90.30 | 90.76 | 90.82 | 90.72 | 91.10 | 90.52 | 90.24 | 90.80 |
| 60 | 90.27 | 90.32 | 90.30 | 90.58 | 90.58 | 90.83 | 90.86 | 90.95 | 90.78 | 90.73 | 90.45 |
| 80 | 90.28 | 90.39 | 90.27 | 90.50 | 90.48 | 90.76 | 90.93 | 90.69 | 90.26 | 90.66 | 90.69 |
| 100 | 90.30 | 90.70 | 90.48 | 90.63 | 90.22 | 90.11 | 90.83 | 90.79 | 90.55 | 90.72 | 90.53 |
| 120 | 90.00 | 90.56 | 90.30 | 90.52 | 90.82 | 90.52 | 90.74 | 90.50 | 90.79 | 90.45 | 90.79 |
| 140 | 90.36 | 90.52 | 90.54 | 90.37 | 90.54 | 91.03 | 90.48 | 90.69 | 90.75 | 90.60 | 90.99 |
| 160 | 90.58 | 90.34 | 90.57 | 90.44 | 90.68 | 90.52 | 90.79 | 90.35 | 90.41 | 90.85 | 90.53 |
| 180 | 90.57 | 90.41 | 90.54 | 89.97 | 90.82 | 90.56 | 90.80 | 90.64 | 90.61 | 90.89 | 90.20 |
| 200 | 90.43 | 90.41 | 90.23 | 90.64 | 90.58 | 90.87 | 90.93 | 90.73 | 90.57 | 90.73 | 91.09 |

**When group-size(gs) is 8, winning tickets (found when gs=1) disappear**

Figure 8: Revisiting the extended lottery ticket experiments of Fig. 1, but with the group size (gs) of 8. As before, $(s, v)$ stands for (epoch drawing the sparse structure, epoch for rewinding). In contrast to Fig. 1, the winning tickets found when gs=1 (green color in Fig. 1) disappear as the group size becomes 8.

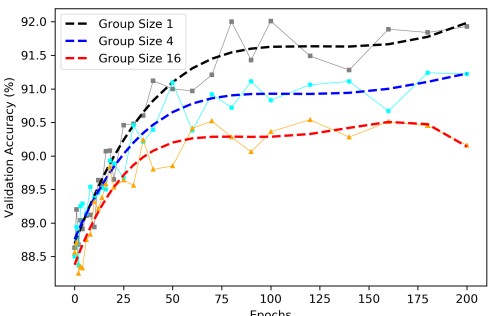

Figure 9: Impact of structured sparsity on the saturation of pruning accuracy. The x-axis corresponds to the epoch when the model is 80% pruned. The dotted lines correspond to the moving average of the accuracy to show trends. The larger the group size, the later the accuracy of the pruned models converges at.

it with Fig. 1, there are two notable points; 1) there is $> 1\%$ accuracy degradation when the group size is increased from 1 to 8, and 2) *when $gs = 8$, the trend that the tickets drawn at epochs $s > 100$ achieve the higher accuracy disappears* (as highlighted with a red rectangle in the figure). (Note that the two figures achieve the similar accuracy when $s < 60$ (colored in yellow), but the tickets with high accuracy (colored in green) can only be seen in Fig. 1.) The former observation makes sense as the more regularization on the sparse weight structure would result in the lower accuracy. But the latter observation is quite surprising; it seems that a distinct behavior of the lottery ticket hypothesis (i.e., the opportunity of finding winning tickets given a proper initialization) is *disrupted* as a group structure is asserted on the sparsity.

We can find a clue on this disrupted behavior by employing the memorization capacity analysis. Table 5 shows the results of memorization capacity experiments on CIFAR10-ResNet20 with an increasing group size $gs = \{1, 2, 4, 8, 16, 32, 64\}$. The memorization capacity degrades significantly as the group size increases. Intuitively, by asserting a large group size, the model's expressivity is degraded and it becomes harder to learn the complex patterns, leading to accuracy loss. Note that we had a similar observation when the tickets are drawn too early (cf.,$s < 60$ in Fig. 1). From this, we can

hypothesize that the winning tickets disappear as the memorization capacity is degraded, which can be observed when the tickets are drawn too early or the group sparsity is forced.

Fig. 9 further demonstrates the impact of structured sparsity on the saturation of pruning accuracy. For different group sizes, the figure shows the accuracy of CIFAR10-ResNet20 80% pruned at varying epochs. It can be observed that the models with the larger group size not only achieve the lower accuracy but also converge at earlier epochs. (E.g., the knee points for $gs = 1$ and $gs = 16$ are around 90 and 50 epochs, respectively.) This experimental result supports our claim that the larger group size enforced in the sparse structure results in the earlier convergence of the pruned models that misses the winning tickets.

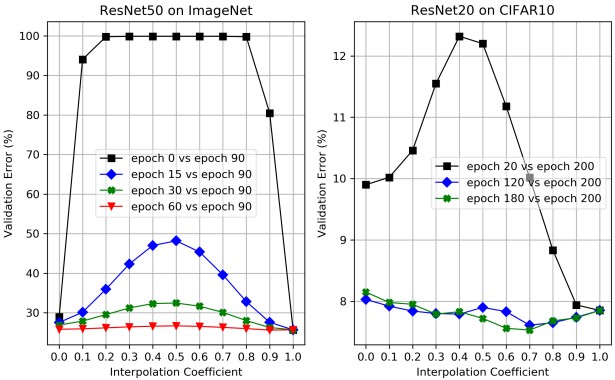

Figure 10: Mode connectivity: Lottery tickets of ImageNet-ResNet50 and CIFAR10-ResNet20 drawn from different epochs are linearly interpolated, then the accuracy is measured for each interpolation coefficient. It can be observed that only the early winning tickets are connected to the winning tickets (drawn at the end of training, i.e., epoch 90 for ImageNet and epoch 200 for CIFAR10).

### B.2 Mode Connectivity of Early Winning Tickets

Mode connectivity is a phenomenon that SGD solutions are connected through paths of approximately equal loss (Draxler et al. (2018)). It provides the perspective of how well the models trained via a proposed method can generalize. In the context of early winning tickets, we for the first time reveal that *the lottery tickets drawn early based on PPR are indeed connected from the one obtained by IMP*, whereas the tickets chosen too early do not.

For the mode connectivity experiments, we draw three lottery tickets - one drawn too early (i.e., a premature ticket, PT), one drawn based on PPR (i.e., an early winning ticket, EWT), and one at the end of the training schedule (i.e., a winning ticket, WT). Note that all the tickets are trained from the same initial weights and with the same retraining schedule. Then we linearly interpolate PT and EWT with WT, and plot the test error for different interpolation coefficient (from 0 to 1).

Fig. 10 shows the results of the mode connectivity experiments for ImageNet and CIFAR10. For both datasets, the linearly interpolated models between PT and WT depict a barrier, indicating that PT and WT are disconnected. Whereas, the linearly interpolated models between EWT and WT show consistently low test error. Note that such linear mode connectivity is observed even when 1) the weights are sparse (i.e., 80% pruning rate) and 2) the mask distance is large (cf., Fig. 3). This intriguing observation counters some observations of prior work; Draxler et al. (2018) claims that it takes a careful search (instead of a simple linear interpolation) to find a pass connecting the modes, and the existence of such a non-linear pass requires enough number of parameters. Our novel observation of early lottery tickets provide a new aspect of sparse model training; the sparse structures mature early in the training, and they converge to a flat local minimum even if its shape is vastly distant. A deeper investigation about the sparse structure of the lottery tickets and the mode connectivity is an interesting future research topic.

## C Early Winning Ticket Algorithms

In this section, we describe the detail algorithms for finding early winning tickets via one-shot and gradual pruning.

---

**Algorithm 1:** Early Winning Ticket Identification with One-Shot Pruning

---

**Require:** Initial weights $\theta_0$, learning rate schedule $LS := \{\eta(n)\}, n = 0, 1, ..., N-1$, pruning ratio $p$, PPR accuracy checking period $T_{PPR}$, PPR accuracy saturation detection threshold $k_{PPR}$, PPR retraining learning rate $\eta_{PPR}$;

Initialize model with $\theta_0$, $t = 0$;

**Step 1: Early winning ticket (EWT) identification by tracking PPR accuracy**

**for** $i = 0; i < N; i{+}{+}$ **do**

    Train the model with learning rate $\eta(i)$ for one epoch;

    **if** $i \% T_{PPR} = 0$ **then**

        Save a copy of the model and optimization state;

        Prune $p\%$ of the model parameters and create a mask $m_{PPR,t}$;

        Train the pruned model $\theta_i \circ m_{PPR}$ for one epoch and obtain the test accuracy $Acc_{PPR,t}$;

        Restore the model and optimization state with the saved copy;

        **if** $Acc_{PPR,t'} <= Acc_{PPR,t'-1}$ *for* $t' = t, t-1, ..., t - k_{PPR} + 1$ **then**

            Set the model parameters to $\theta_i \circ m_{PPR,t}$;

            break;

        **end**

        $t{+}{+}$;

    **end**

**end**

**Step 2: Retraining EWT with the same $LS$**

**for** $i = 0; i < N; i{+}{+}$ **do**

    Train the pruned model with learning rate $\eta(i)$ for one epoch;

**end**

---

---

**Algorithm 2:** Early Winning Ticket Identification with Gradual Pruning

---

**Require:** Initial weights $\theta_0$, learning rate schedule $LS := \{\eta(n)\}, n = 0, 1, ..., N-1$, pruning ratio $p$, PPR accuracy checking period $T_{PPR}$, PPR accuracy saturation detection threshold $k_{PPR}$, PPR retraining learning rate $\eta_{PPR}$, gradual pruning period $T_{gr}$;

Initialize model with $\theta_0$, $t = 0$;

**Step 1: Early winning ticket (EWT) identification by tracking PPR accuracy**

**for** $i = 0; i < N; i{+}{+}$ **do**

    Train the model with learning rate $\eta(i)$ for one epoch;

    **if** $i \% T_{PPR} = 0$ **then**

        Save a copy of the model and optimization state;

        Prune $p\%$ of the model parameters and create a mask $m_{PPR,t}$;

        Train the pruned model $\theta_i \circ m_{PPR}$ for one epoch and obtain the test accuracy $Acc_{PPR,t}$;

        Restore the model and optimization state with the saved copy;

        **if** $Acc_{PPR,t'} <= Acc_{PPR,t'-1}$ *for* $t' = t, t-1, ..., t - k_{PPR} + 1$ **then**

            break;

        **end**

        $t{+}{+}$;

    **end**

**end**

**Step 2: Gradual Pruning to refine the EWT structure**

**for** $i = 0; i < T_{gr}; i{+}{+}$ **do**

    Prune model gradually toward the pruning ratio $p$ according to Eq. (6) and create a mask $m_{gr,i}$;

    Train the pruned model $\theta_{gr,i} \circ m_{gr,i}$ for one epoch;

**end**

**Step 3: Retraining EWT with the same $LS$**

**for** $i = 0; i < N; i{+}{+}$ **do**

    Train the pruned model with learning rate $\eta(i)$ for one epoch;

**end**

---

