# OpenReview forum: "The Sooner The Better: Investigating Structure of Early Winning Lottery Tickets"
_ICLR.cc/2020/Conference — Reject_

### Official Review · AnonReviewer1 · 2019-10-22
**Official Blind Review #1**

**Rating:** 3

**Review:**

This paper attempts an in depth study of the lottery ticket hypothesis. The lottery ticket hypothesis holds that sparse sub-networks exist inside dense large models and that the sparse sub-networks achieve at least as good an accuracy as the underlying large model. These sub-networks are discovered by training and iteratively pruning the dense model. This paper investigates the epoch at which pruning should occur as well as the epoch at which weights should be rewound when retraining. Then, the authors conduct experiments with different pruning strategies (one-shot vs. gradual) in an attempt to find such sparse models (or "winning tickets") earlier than they otherwise would have been found.

The experiments conducted by the authors seem to be very extensive, and I think the paper contains useful data to have for researchers interested in better understanding the lottery ticket hypothesis. However, my main issue is with both the originality and significance of this work. This paper gives evidence that winning tickets may be found "early," although their notion of early still involves quite a lot of training.

Although the paper is interested in addressing the structure of the winning tickets, I really didn't find any of the discussion of structure to give much insight into the lottery ticket hypothesis. Most of the section focuses on analyzing weight magnitude, though I was hoping for something more about the actual structure of the sparse subnetwork -- especially given the title of the paper. Figure 3 is notable, showing that different winning tickets (parameterized by different prune and rewind epochs) can have a large Hamming distance between them. This is very interesting, and I wish the authors had more to say. How is this affected by different initializations? Are these solutions connected on a loss landscape? Is there something invariant about the sparse architecture after symmetries are taken into account? It's not clear to me that Hamming distance alone is enough.

In conclusion, the paper presents a set of nice experiments, but doesn't really shed too much additional light on the scientific nature of the lottery ticket hypothesis.

**Experience Assessment:**

I have read many papers in this area.

**Review Assessment: Checking Correctness Of Derivations And Theory:**

N/A

**Review Assessment: Checking Correctness Of Experiments:**

I assessed the sensibility of the experiments.

**Review Assessment: Thoroughness In Paper Reading:**

I read the paper at least twice and used my best judgement in assessing the paper.

---

> ### Author Response · Authors · 2019-11-14
> **Response to the reviewer's comments**
>
> The authors appreciate the reviewer's constructive and insightful comments and suggestions. We carefully reviewed all the comments you raised and revised the manuscripts to address them. Here we provide our answers to your questions/comments.
>
> Q1) Originality and significance of discovery about our early winning tickets
> A1) We agree with the reviewer that the early winning tickets we propose still require a noticeable amount of training epochs to be found. However, our novel observation that the winning tickets can be drawn “early but not too early” sheds light on a deeper understanding of the behavior of training sparse models. In particular, we demonstrate in the paper that it takes a significant number of epochs to find a proper sparse structure that would result in achieving comparable accuracy. We are among the first to connect the memorization capacity and the learning dynamics (capturing simple or complex patterns with large or low learning rates, respectively) to quantitatively understand the training behavior of sparse models, which leads to a practical method to identify the winning tickets early. Employing this method results in not only the strong empirical supports of our claims but also the practical savings of training effort.
>
> Q2) More investigation about the actual structure of the sparse subnetworks
> A2) In the initial submission, we did not add the discussion on the various aspects of the structure of our sparse models due to the page limit. Thanks to the reviewer, we realize that it can be of great interest to the readers. Thus, we now include two sections in Appendix to discuss about interesting characteristics of the structure of the sparse models and the lottery tickets hypothesis (LTH): the impact of structured sparsity on LTH (Appendix B.1) and the mode connectivity of early winning tickets (Appendix B.2, as greatly motivated by the insightful suggestions from the reviewer).
>
> The former provides an interesting observation about the impact of the shape of the sparse structure – i.e., what if we force the structure in the sparsity patterns, like grouping weights along the channel when they are pruned? This investigation has a relationship with a well-known observation from the popular “channel pruning” papers where the highest pruning rate achieved by it (i.e., group size=channel) is typically much lower than the pruning rate of the individual weight pruning (i.e., group size=1). In Appendix B.1., we provide interesting reasoning behind this intriguing discrepancy that the distinct behavior of the lottery ticket hypothesis (i.e., the opportunity of finding winning tickets given a proper initialization) is disrupted as a group structure is asserted on the sparsity. We reveal that the impact of such disruption is similar to what we observe when the tickets are drawn too early, opening up a new research direction relating to the structured sparsity and LTH.
>
> The second discussion topic is to combine the analysis framework of “mode connectivity” with the lottery ticket hypothesis. We for the first time reveal that the early winning tickets found by our method are connected in the loss landscape by a simple linear interpolation, for both CIFAR10 and ImageNet cases, implying that the loss landscape converged from the sparse structures (= lottery tickets) is flat as long as they are found “early but not too early”. We demonstrate that as the lottery tickets are found based on our PPR metric, the modes from these tickets are connected. Our novel observation of early lottery tickets provides a new aspect of sparse model training; the sparse structures mature early in the training and they converge to a flat local minimum even if their shape is vastly different (which is demonstrated in terms of mask distance).
>
> Please see the detail discussion in Appendix B. We thank the reviewer for leading us to include such interesting discussions.

---

### Official Review · AnonReviewer3 · 2019-10-23
**Official Blind Review #3**

**Rating:** 3

**Review:**

Overview:

The paper is dedicated to conducting an in-depth investigation of the structure of winning lottery tickets. The author provides supporting evidence for the structure of the early winning tickets: 1) lottery tickets emerge when the weight magnitude of a model saturates with SGD optimization. 2) pruning before model saturation may result in accuracy degradation. In the experiment part, they employ the memorization capacity analysis and discover the early wining tickets without expensive iterative pruning. The author also conducts extensive experiments with various ResNet architectures on both CIFAR 10 and ImageNet, achieving state-of-the-art results with only 1/5 of the total epochs for iterative pruning.

Strength Bullets:

1. The experiment organization is complete and convincing. Especially for the figure, it not only clearly shows that lottery tickets emerge much earlier before full training ends, but also shows the effect of rewinding.
2. The author not reveal interesting observations, but also provides useful guidance. It is a complete logic chain that is also aligned with my intuition. For example, first, the author discusses the memorization capacity of different pruned models at different epochs. Then, they introduce a reasonable gradual pruning technique. Finally, they conduct experiments to confirm it.
3.  The early winning tickets in this paper achieve state-of-the-art results with only 1/5 of the total epochs for iterative pruning.

Weakness Bullets:

1. For lottery tickets, especially for early winning tickets, I think there is a lot of randomnesses. Thus, for the plot like figure 2, figure 5, they need to contain an error bar and the curve should be the average of tens of experiments. It will be more convincing if it decouples the randomness from the real patterns.
2. The description and organization of section 4 need to be more clear. For example, an algorithm pseudo code will definitely give readers a much more clear understanding of the early winning tickets finding strategy.



Question the authors don't answer which confuses me more:
Need more convincing analysis about the indicator - Hamming Distance

Just as the comment I posted after Review 1, we would like to see more analysis about Hamming Distance between different winning tickets. The author mentioned in the following way:

''However, as we demonstrate in Fig. 3 of our paper, the mask-distance does not well characterize the winning tickets. E.g., the lottery tickets drawn at Epoch 120 and 200 have a mask distance of 0.082 in Fig. 3, which is much larger than the mask distance between Epoch 180 and 200. Whereas, all three tickets achieve comparably high accuracy as shown in Fig. 1, implying a shallow correlation between the accuracy and the mask distance''

As far as I know, this observation only tells us that the structure of the lottery tickets changes, which are drawn from 120 epochs to 180 epochs (although the maintain a similar accuracy). However, we can not conclude that the mask distance is not a reliable measure. As mention by the [EB] paper provided by the authors, you can use the rate of the distance change to indicate the early winning tickets. In this way, we can find winning tickets much earlier than authors' work.

To better address this question, I suppose the authors need to provide more analysis of the mask distance indicator. I think all three reviewers would like to see the results. A good indicator for early winning tickets is very important, otherwise authors' notion of early still involves quite a lot of training.

Recommendation:

Due to the unsolved important question, here is a weak reject.

**Experience Assessment:**

I have read many papers in this area.

**Review Assessment: Checking Correctness Of Derivations And Theory:**

I carefully checked the derivations and theory.

**Review Assessment: Checking Correctness Of Experiments:**

I carefully checked the experiments.

**Review Assessment: Thoroughness In Paper Reading:**

I read the paper thoroughly.

---

> ### Author Response · Authors · 2019-11-14
> **Response to the reviewer's comments**
>
> The authors appreciate the reviewer's insightful comments and suggestions. Here we provide our answers to the reviewer’s questions/comments. Also, please note that we have updated the manuscript to add more experimental results as well as an interesting discussion on various aspects of winning lottery tickets in the Appendix.
>
> Q1) Need to better account for the randomness across the experiments of early lottery tickets.
> A1) As per the reviewer's sensible comments, we have updated the selected experimental results to account for the impact of randomness when the overall performance of the proposed method is measured. Specifically, Fig. 5 and Table 2 now represent the average and deviation of accuracy among 10 independent runs; still, the same trend is clearly observed. For those experiments that require the consecutive pruning, such as Fig. 2, it is critical to control randomness across the runs at different epochs. Thus, we carefully control the randomness of the experiments (e.g., by fixing the random seed) across the experiments so that each run can employ the same experimental setting. Detail descriptions indicating such control have been added to the manuscript.
>
> Q2) Clarify Sec. 4 (Add a clear description of the algorithms)
> A2) Thanks to the reviewer for improving the clarity of our paper, In Appendix C, we have added the two algorithms to find the winning lottery tickets via one-shot and gradual pruning. The Appendix also additionally contains an in-depth analysis of structured pruning results that can provide a clearer understanding of the algorithms described in Section 4. Furthermore, we've updated and re-written the manuscript and figure (4 and 5) to show a more clear explanation of the sub-network structures obtained by the early lottery ticket algorithm.

---

### Official Review · AnonReviewer2 · 2019-10-23
**Official Blind Review #2**

**Rating:** 6

**Review:**

This paper carefully observes the behavior of weight magnitudes during training, finding the is a stage of saturation that is closely related to the winning lottery tickets drawing. Based on this observation the authors hypothesize that we can draw lottery tickets early but too early pruning can irreversibly hurt the learning capability for complex pattern. To remedy this and draw the tickets as soon as possible, the authors propose to adopt gradual pruning, which 1) can start early without hurting the learning capability too much; 2) avoid computation-heavy iterative pruning in previous works.

Questions:

1. Overall I am very happy with the interesting observations and analysis of the dynamics of weight magnitudes and how it can be related to the early winning lottery tickets drawing. But how valuable is it for practical use? In practice, we cannot know in advance when to start (gradual) pruning.

2. In Fig.1, what does it mean if we perform weight-magnitude based pruning at 10th epoch but rewind the weight to the 20th epoch? Is there a baseline network that is normally trained straight to the end and to which we rewind all pruned models?

3. I am not quite convinced by the experiment of Fig. 4 and argument at the bottom half of page 5. I buy the intuition that pruning too early might irreversibly hurt the capability of learning complex pattern. But I have trouble understanding how the experiment of Fig. 4 supports this intuition. The curve of retraining with smaller LR (0.01) has the save trend as the baseline and retraining with larger LR (0.1). Retraining only one epoch can hardly convince me of its relationship with learning capability. Also, for the experiment in Fig. 5, to validate the proposed hypothesis, it's more valuable to provide results around the claimed turning point, i.e. around 100 epoch instead of suddenly jumping from 20 epoch to 120 epoch.

4. In Tab. 2, the results of ResNet56 with gradual pruning is not presented. In Tab. 3, the results of ResNet50 with one-shot pruning is not presented. It would be better to have these results for clear comparison.

Overall, I love the empirical observation of weight magnitudes and think it would help the community to understand lottery tickets and training process of deep models.

Update:
The response from the authors addressed some of my questions and more experiments were added per my suggestions. However, also considering the authors' response to R#1 and R#3, I don't think it's strong enough for me to raise my score. Therefore I will keep my current score.

**Experience Assessment:**

I have read many papers in this area.

**Review Assessment: Checking Correctness Of Derivations And Theory:**

I carefully checked the derivations and theory.

**Review Assessment: Checking Correctness Of Experiments:**

I carefully checked the experiments.

**Review Assessment: Thoroughness In Paper Reading:**

I read the paper thoroughly.

---

> ### Author Response · Authors · 2019-11-14
> **Response to the reviewer's comments**
>
> The authors appreciate the reviewer's insightful comments and suggestions. We carefully reviewed all the comments you raised and revised the manuscripts to address them. Here we provide our answers to your questions/comments.
>
> Q1) Practical use of the proposed algorithms for finding early lottery tickets.
> A1) The dynamics of weight magnitudes give us intuition when we can find a pruned structure to achieve reliable accuracy. In practice, we propose to use the pseudo-pruned-then-retrained (PPR) method to determine when to apply pruning. As shown in Fig. 6, this enables early termination of the costly iterative pruning procedure and kick-start of (more efficient) training the sparse model. We now included the detailed algorithms of finding the early winning tickets for both one-shot and gradual pruning in Appendix C., in order to clearly show how we decide when to start the one-shot and gradual pruning based on PPR accuracy. In Sec. 4.3, we demonstrate that the proposed algorithms can achieve significant savings in terms of the number of epochs for training a sparse model. With a sparsity aware accelerator hardware, we can further expect to gain speedup from the reduced FLOPS for training sparse models.
>
> Q2) Clarification of experiments in Fig. 1
> A2) Previous studies suggested that to successfully train a lottery ticket, both the pruned structure and the rewinding point are important. To further understand the relationship between the state of the weight (magnitude) and the structure (indices of unpruned weights), we tested various combinations of points when to prune and when to rewind. For example, pruning at 10th epoch but rewind to the 20th epoch, means that we obtain the pruned structure (a subset of the model) at epoch10, but keep training the full model until epoch 20, then use the weights of this subset at epoch 20 to initialize the pruned structure to further finish the training. The results (Figure 1) shows that, once we get an early ticket from a matured model, this "forward" rewinding has no impact on the final accuracy, while normal "backward" rewinding cannot be too early (>10 epochs).
>
> Q3) Clarify the experiment in Fig. 4 and 5.
> A3) Thanks for pointing out that Fig. 4 needs more clarification; here we attempt to provide a clearer explanation. The experiment in Fig. 4 is motivated by the claims from Li et al. (2019) that the model capacity to capture simple or complex patterns is highly related to the learning rate. We observed that when the pseudo-pruned model retrained with the large learning rate (i.e., lr=0.1), the accuracy curve of the pruned model coincides with the baseline model with the fixed learning rate of 0.1. This implies that across all the pruning epochs the pruned model maintains the capability of capturing the simple patterns if the large learning rate is used. Whereas, if the pseudo-pruned model is retrained with the lower learning rate (i.e., lr=0.01), interestingly the model’s accuracy improves over the pruning epochs. From this, we can conjecture that the pruned model recovers its capability of capturing complex patterns as it is pruned at the later epochs. In the case of CIFAR10-ResNet20, if we zoom in, we can further observe that such recovery of accuracy saturates at around epoch 100. In the case of ImageNet-ResNet18, as shown in Fig. 6, this saturation (indicating the early winning tickets) can be more vividly observed. We further validate this conjecture with the memorization capacity in Fig. 5.
> Given that the retraining of the pseudo-pruned model with the low learning rate reveals the early winning tickets, a practical question is to determine how many epochs to retrain. Surprisingly, we found that just one epoch of retraining is enough to reveal the winning tickets. In fact, the use of validation accuracy as a proxy for evaluating the quality of the pruned models is not new; AMC (He et al. 2019) employed validation accuracy of a pruned model without retraining to estimate the quality of the pruned model for tuning the reinforcement learning agent. But we empirically found that no-retraining results in the accuracy as low as only ~30%, whereas one-epoch retraining provides a much more robust estimation of the true validation accuracy, as shown in Fig. 4.
> Also, we have updated Fig. 5 to separate the plots to better visualize the trend and to account for randomness in the experiments by representing the average and min/max of the accuracy across 10 runs.
>
> Q4) Add experimental results
> A4) Thanks to the reviewer for the constructive suggestion. We have updated Table 2 and 3 to further include CIFAR10-ResNet56 for gradual pruning and ImageNet-ResNet50 for one-shot pruning, respectively. As the reviewer pointed out, these complete sets of pruning experiments will greatly strengthen our claims on the early winning tickets.

---

> > ### Author Response · Authors · 2019-11-14
> > **Additional interesting discussion added in the Appendix of the manuscript**
> >
> >  Also, please note that we have updated the manuscript to add more experimental results as well as an interesting discussion on various aspects of winning lottery tickets in the Appendix.

---

### Author Response · Authors · 2019-11-14
**Comparing our work with a parallel work submitted to ICLR2020**

We notice that there is a parallel work submitted to ICLR2020 about finding winning lottery tickets early: “Drawing Early-Bird Tickets” ([EB], https://openreview.net/forum?id=BJxsrgStvr). The authors of [EB] claim that the winning tickets (via channel pruning) can be found as early as only a few tens of epochs with a mask distance, which counters with our claims (i.e., the early winning tickets can be found early but not too early, and the mask distance is not a distinctive measure for identifying the winning tickets). In this comment, therefore, we would like to provide the reasons for such discrepancy.

Let us begin with several serious concerns in [EB].

1) It seems that the accuracy results of the prior work used in [EB] are significantly underrated. The authors of [EB] mentioned that they share many experimental settings with “Network Slimming” paper (Liu et al. 2017, NS), but the accuracy of NS on VGG16-CIFAR10 reported in this paper (= 90.4% at 70% pruning) is much lower than what is reported in the original NS paper (= 93.8% at 70% pruning). The similar concerns can be found in [EB] for the other reference works -- Lottery ticket hypothesis (Frankle & Carbin 2018, LTH) and Single one-shot pruning (Lee et al. 2018, SNIP). If [EB] shares similar experimental settings with NS, this big gap in pruning performance is very confusing.

2) It seems that the characteristics of the winning tickets claimed in [EB] are vastly different from the winning tickets in the original LTH paper. In fact, many papers discussing LTH share the same underlying assumption that the target pruning rate should reach the state-of-the-art results; e.g., NS achieved 70% channel pruning with matching accuracy. Strangely, the authors of [EB] presented a very trivial pruning performance in their demonstration: 10% channel pruning for VGG16-CIFAR10 (Fig. 1 in [EB]). Beyond this pruning rate, the tickets they draw do not seem to match the baseline accuracy (i.e., they are “failed” tickets).

3) A more disturbing observation in [EB] is that the “late” tickets behave so badly. E.g., in the case of CIFAR100 networks, the accuracy goes down significantly as the epoch for drawing a ticket increases. This phenomenon seriously counters with the observation of the original LTH paper and many others, where the winning tickets are usually drawn at the end of the training epochs (thus the “late” tickets). The authors refer to (Achille et al., 2019) for a handwavy explanation about this phenomenon, but (Achille et al., 2019) do not explain why the “late” tickets behave badly.

Now we clarify the discrepancy in [EB] and provide the reasonings.

First, [EB] claims that they can find winning lottery tickets (via channel pruning) as early as a few tens of epochs for CIFAR10 networks (e.g., 14 epochs for PreResNet, [EB, Fig. 3]). This counters with our claims – it would require at least 100 epochs to find the highly accurate (= winning) lottery tickets. We conjecture that this discrepancy is because [EB] attempts to connect LTH with the structured pruning without careful justification. In fact, we provide several evidences on why finding “structured” winning tickets too early is not possible. In Sec. 3 of our paper, we explain that the early phase of training (1~100 epochs) is important to establish the reliable structure of weights, particularly when we target the state-of-the-art pruning rate (e.g., >80% pruning rate for element-wise pruning on ResNet20-CIFAR10). In Appendix B.1, we further shed light on understanding the relationship between the structured pruning and the lottery ticket hypothesis (LTH). We demonstrate that the larger group size is asserted in pruning, the more the behavior of LTH disappears. These counter-evidences indicate that a naïve connection between LTH and the channel pruning in [EB] would require careful justification.

Second, [EB] proposes to use mask-distance as a metric for finding winning lottery tickets. However, as we demonstrate in Fig. 3 of our paper, the mask-distance does not well characterize the winning tickets. E.g., the lottery tickets drawn at Epoch 120 and 200 have a mask distance of 0.082 in Fig. 3, which is much larger than the mask distance between Epoch 180 and 200. Whereas, all three tickets achieve comparably high accuracy as shown in Fig. 1, implying a shallow correlation between the accuracy and the mask distance. We can further compare the mask distance with the mode connectivity investigated in Appendix B.2; we can observe that the tickets drawn at epoch 120 and 180 are connected with the ticket at epoch 200. From these observations, we can conclude that the mask distance may not be a reliable measure for finding early winning tickets.

---

> ### Author Response · Authors · 2019-11-14
> **Comparison continued.**
>
>
> We acknowledge that [EB] has its own contribution to finding a direct connection between the phenomena of early lottery tickets and the energy savings in training, which might be of great interest to many readers. Therefore, we believe that the in-depth analysis, insights and strong supporting evidence provided in our paper can greatly increase the clarity of the claims in [EB], which are in part made with the restricted empirical justification (e.g., the target pruning rate is low and the “late” lottery tickets achieve very poor accuracy).

---

### Decision · Program_Chairs · 2019-12-19

**Decision:**

Reject

**Comment:**

This paper does extensive experiments to understand the lottery ticket hypothesis. The lottery ticket hypothesis is that there exist sparse sub-networks inside dense large models that achieve as good accuracy as the original model. The reviewers have issues with the novelty and significance of these experiments. They felt that it didn't shed new scientific light. They felt that epochs needed to do early detection was still expensive. I recommend doing further studies and submitting it to another venue.